# Use of convolutional neural networks for the automatic segmentation of total retinal and choroidal thickness in OCT images

**David Alonso-Caneiro, Scott A. Read, Jared Hamwood, Stephen J. Vincent, Michael J. Collins**
Contact Lens and Visual Optics Laboratory
School of Optometry and Vision Science
Queensland University of Technology
Brisbane, Queensland, Australia
`d.alonsocaneiro@qut.edu.au`

## Abstract

The assessment of total retinal and choroidal thickness from optical coherence tomography (OCT) images is an important clinical and research task. These thickness measures and their changes represent a fundamental metric extracted from OCT data, since they provide valuable information regarding the eye's normal anatomy and physiology. Changes in thickness are associated with natural eye development, the progression of various eye diseases, and the development of refractive error. Manual analysis of OCT images is time-consuming and not feasible for large datasets of images. Thus, the development of reliable and accurate methods to automatically segment tissue boundaries in OCT images is fundamental. In this paper, convolutional neural networks (CNNs) are used to calculate the probability of boundary locations in OCT images. The CNN, trained using image patches centred around the boundary of interest, provides a per-layer probability map that marks the most likely predicted location of the boundaries. This map is subsequently traced using a graph-search approach to segment the boundaries. The effect of patch size, network architecture and input image pre-processing on the CNN performance and subsequent layer segmentation is presented. The results are compared with manual image segmentation as well as a fully convolutional network. This work may support the future development of CNN methods for automated OCT boundary segmentation.

## 1 Introduction

Optical coherence tomography (OCT) has become one of the key methodologies for imaging the eye [1,2]. This technology provides high-resolution cross-sectional images of the eye's tissues, which are commonly used in both clinical practice and research to better understand the eye in both health and disease. OCT images of the posterior segment of the eye provide a detailed view of the complex anatomy of this region of the eye. Total retinal and choroidal thickness are amongst the most common metrics extracted from OCT images, this is especially important in cases of pathology where layer morphology may be altered due to the disease.

Manual delineation of the layers in OCT images is time consuming and, therefore, impractical for the large number of B-scan images acquired in dense scanning protocols often used in clinical settings and for large scale population studies. A number of methods have been proposed to segment retinal and choroidal layers. While earlier methods are based on more standard image analysis procedures [3,4], recently a number of segmentation methods make use of more sophisticated deep learning algorithms. For retinal segmentation a number of methods have been proposed. Lang et al. [5] used a

1st Conference on Medical Imaging with Deep Learning (MIDL 2018), Amsterdam, The Netherlands.

Random Forest Classifier (RFC) to classify features around a pixel into layer classes. Ben-Cohen et al. [6] and Venhuizen et al. [7] utilized the U-net fully convolutional architecture to identify retinal boundaries. Roy et al. [8] created an encoder-decoder framework similar to U-net, called ReLayNet, to segment the image into retinal tissue as well as intra-layer fluid. To date, the segmentation of the choroid using deep learning methods has not received as much attention. Shiu [9] proposed a multi-scale CNN networks, which concatenates three networks in different scales: coarse, middle and fine. This provides a map of probability that is later on segmented using a graph search approach.

Recently, Fang et al. [10] proposed a convolutional neural network (CNN) and graph-search method (termed as CNN-GS) for the automatic segmentation of nine layer boundaries in OCT retinal images of patients with non-exudative age-related macular degeneration. We have recently presented an in depth analysis of this methodology to better understand the effect of using different patch sizes on segmentation performance for retinal layer segmentation [11]. In this paper, we extend this approach to the analysis of the choroid in OCT images, to better understand the application of different CNN architectures for the segmentation of this layer. The total retina is included as it provides a baseline for comparison. We compare the findings with fully convolution networks (ReLayNet) and we also explore the impact of different inputs on the segmentation performance. The findings of this work may inform future development of deep learning strategies in OCT imaging analysis.

## 2 Methods

### 2.1 OCT dataset: Demographics and details

A retrospective dataset of OCT images was used for this study [12], obtained from a longitudinal study examining macular retinal layer thickness in childhood involving 101 children ($13.1 \pm 1.4$ years) with a range of refractive errors. All participants had normal vision in both eyes and no history or evidence of ocular disease, injury or surgery. The study was approved by the Queensland University of Technology human research ethics committee and all study procedures followed the tenets of the Declaration of Helsinki.

High-resolution cross-sectional retinal images were collected using the Heidelberg Spectralis (Heidelberg Engineering, Heidelberg, Germany) SD-OCT instrument. The captured images have pixel dimensions of 496 x 1,536 pixels and the resolution is 3.9 x 5.7 μm per pixel respectively (deep x wide). The OCT images were manual segmented by an expert image grader and analyzed using custom written software to segment the three boundaries of interest including: the outer boundary of the retinal pigment epithelium (RPE), the inner boundary of the inner limiting membrane (ILM) and the chorio-scleral interface (CSI)(Figure 1).

### 2.2 Overview of image processing methods

The method used for the segmentation of OCT images follows a similar procedure to that used by Fang et al [10]. Briefly, this method uses the CNN to compute a probability map that indicates the likelihood of a boundary lying at any given pixel. This probability map is then used as an input to the graph-search method for layer segmentation. The graph-search method is used to find the highest weighted probability path from one side of the scan to the other side for each boundary using Dijkstra's algorithm. All training and testing of the networks was computed on an Nvidia Titan Xp using MATLAB r2017a and VLfeat's MatConvNet library [13].

In summary, the first step involves the training of the CNN network, using a dataset with three labelled boundaries of interest. The second step involves the testing of the trained network performance to produce probability maps and then subsequent graph-search to trace these boundary locations. The Fang method adapted the Cifar-CNN architecture, which uses a 33 x 33 pixel input for retinal OCT image layer segmentation. Recently, we explored the effect that patch size has on boundary detection of the retinal layers [11]. The network architecture was modified to take into account the distribution of the information within the OCT image and adapts the patch size to better capture the richness of the information and improve image segmentation performance. Table 1 provides the details for the different networks tested in this work.

## 2.3 Analysis: Performance assessment

A total of 138 OCT images from 71 subjects were used to form the training set, with 28 of these scans used as validation samples during training. Training was done by stochastic gradient descent with momentum, in batches of 1024 randomly chosen samples at a time. To evaluate the performance of the network, 102 OCT images from 17 subjects were used to create a test dataset. All subjects in the test dataset were different to those in the training set, including validation subjects.

Table 1. Architecture of the three different CNNs used in this work. For comparison, the 33 x 33 network proposed by [10] was included as a baseline. The two other proposed CNNs extend the original input size to take advantage of the distribution of the information within the OCT image. To facilitate comparisons across networks, empty rows (indicated with a dash) have been included in the table. The number in round brackets indicates the number of filters, number in angled brackets indicates stride, and number in curly brackets indicates padding.

| 33x33 greyscale patch | 65x65 greyscale patch | 31x61 greyscale patch |
|---|---|---|
| 5x5 Convolutional (32) <1> 2 | | 5x3 Convolutional (32) <1> 0 |
| 3x3 Max Pooling <2> 0 1 0 1 | | - |
| ReLU | | ReLU |
| - | 5x5 Convolutional (32) <1> 2 | 5x3 Convolutional (32) <1> 0 |
| - | ReLU | ReLU |
| - | 3x3 Avg. Pooling <2> 0 1 0 1 | - |
| 5x5 Convolutional (32) <1> 0 | | 5x3 Convolutional (64) <1> 0 |
| ReLU | | - |
| 3x3 Avg. Pooling <2> 0 1 0 1 | | 2x2 Avg. Pooling <2> 1 |
| 5x5 Convolutional (64) <1> 0 | | 5x3 Convolutional (64) <1> 0 |
| ReLU | | ReLU |
| - | | 5x3 Convolutional (128) <1> 0 |
| 3x3 Avg. Pooling <2> 0 1 0 1 | | 2x2 Avg. Pooling <2> 1 |
| 4x4 Fully Connected (64) <1> 0 | | 9x5 Fully Connected (128) <1> 0 |
| ReLU | | - |
| 1x1 Fully Connected (8) | | 1x1 Fully Connected (8) |

OCT contrast enhancement techniques [14], also known as attenuation coefficients [15], are commonly used to improve the boundary visibility particularly for the posterior choroid and to remove shadows caused by the retinal blood vessels. The technique works under the assumption that local backscattering can be related to that of the corresponding attenuation, and therefore can be compensated. The network architecture that provided the best performance, from the three presented in Table1, was also tested with three different network-input options; (i) standard intensity, (ii) attenuation coefficient equivalent and (iii) a combination of both (dual).

To further understand the performance of the methods in the context of previously proposed techniques, the best architecture was compared with the ReLayNet [8], created using an encoder-decoder framework similar to U-net. The network provides a pixel-wise classification of the data (region classification), to extract the position of the boundary and edge detection was applied to the classification area provided by ReLayNet followed by a graph-search to trace the boundary.

To assess the overall performance and the impact of the network architecture, the difference between the predicted boundary location for each network architecture and the true boundary location (as defined by the manual analysis by an expert image grader) was calculated using both the mean error and the mean absolute error, with the standard deviation (SD) of both errors also included.

## 3 Results

**Effect of CNN architecture on the performance:** Table 2 presents the effect of the CNN architecture on the performance. The mean error for the ILM and RPE is low across the different networks, although the rectangular 31 x 61 network provides the best performance in terms of both mean error and SD. The choroidal boundary however, performed poorly using the square networks, with superior performance observed using the rectangular network. Figure 1 illustrates the per-boundary probability

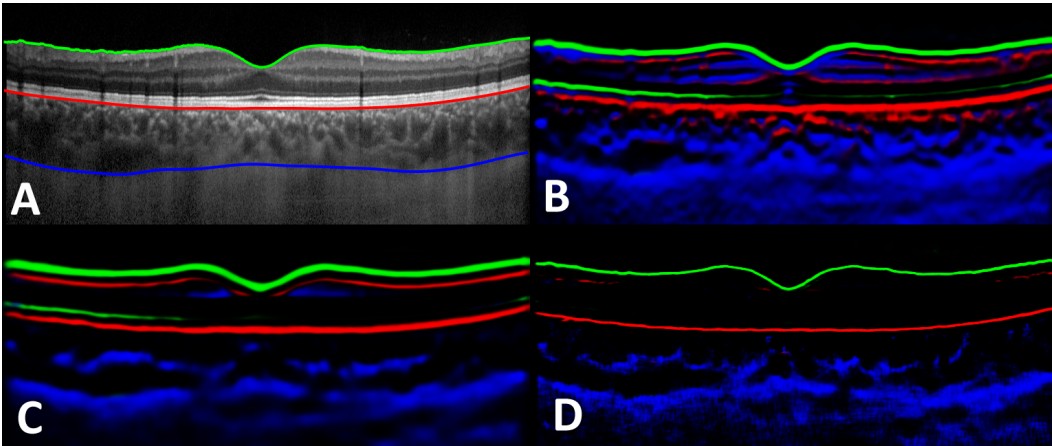

Figure 1: Figure 1. Example B-scan with manual segmentation of the layers of interest (A) and the corresponding probability maps for the three layers (ILM in green, RPE in red and CSI in blue). Each colour indicates a high probability of a boundary to be present in that location. The probability images show the effect on performance while training the CNN with different networks (B is the 33x33, C is the 65x65 and D is the 31x61).

for the different architectures. The rectangular network provides well-defined probability for all the boundaries, which corresponds with the results in Table 2. The rectangular network was therefore used for the remaining evaluations of the methods.

Table 2. The difference in boundary position for each network architecture and the manual observer for the entire dataset. The results are reported as the mean value and (standard deviation) in pixel units. MAE = mean absolute error, ME = mean error.

|  | 33x33 | | 65x65 | | 31x61 | |
|---|---|---|---|---|---|---|
|  | MAE | ME | MAE | ME | MAE | ME |
| ILM | 1.34 (5.51) | -0.66 (5.63) | 1.64 (4.68) | -0.70 (4.92) | 0.62 (2.08) | -0.15 (2.16) |
| RPE | 1.40 (1.64) | 0.14 (2.15) | 1.02 (2.19) | -0.07 (2.41) | 0.52 (0.53) | 0.01 (0.74) |
| CSI | 19.26 (42.38) | 18.05 (42.90) | 10.47 (32.32) | -2.44 (33.87) | 3.24 (4.83) | 0.95 (5.74) |

**Effect of input image on the performance:** Table 3 shows the performance for the different inputs using a rectangular patch size of 31x61 pixels to train the network. The three different network-inputs appear to only have a minor effect on the boundary error, with the dual input yielding a slightly smaller mean error across the different metrics for each boundary. However, these differences are relatively minor across the different inputs for the tested data. Thus, the visual enhancement of the image does not seem to result in a substantial segmentation improvement using the proposed method.

Table 3. The difference in boundary position for each network-input and the manual observer for the entire dataset. The results are reported as the mean value and (standard deviation) in pixel units. MAE = mean absolute error, ME = mean error.

|  | Intensity | | Attenuation Coefficient | | Dual | |
|---|---|---|---|---|---|---|
|  | MAE | ME | MAE | ME | MAE | ME |
| ILM | 0.62 (2.08) | -0.15 (2.16) | 0.56 (0.92) | -0.07 (1.08) | 0.54 (0.80) | -0.00 (0.96) |
| RPE | 0.52 (0.53) | 0.01 (0.74) | 0.58 (0.60) | -0.14 (0.82) | 0.53 (0.54) | 0.02 (0.76) |
| CSI | 3.24 (4.83) | 0.95 (5.74) | 3.25 (4.69) | 1.22 (5.58) | 3.18 (5.38) | 0.64 (6.22) |

**Fully CNN versus CNN-GS:** Table 4 compares the performance of the proposed rectangular network versus a fully CNN (ReLayNet). The performance for the well-defined retinal boundaries is similar across the networks. However, for the CSI boundary the CNN-GS outperforms the ReLayNet that shows a higher standard deviation error. It is worth noting that the ReLayNet provides a per-pixel

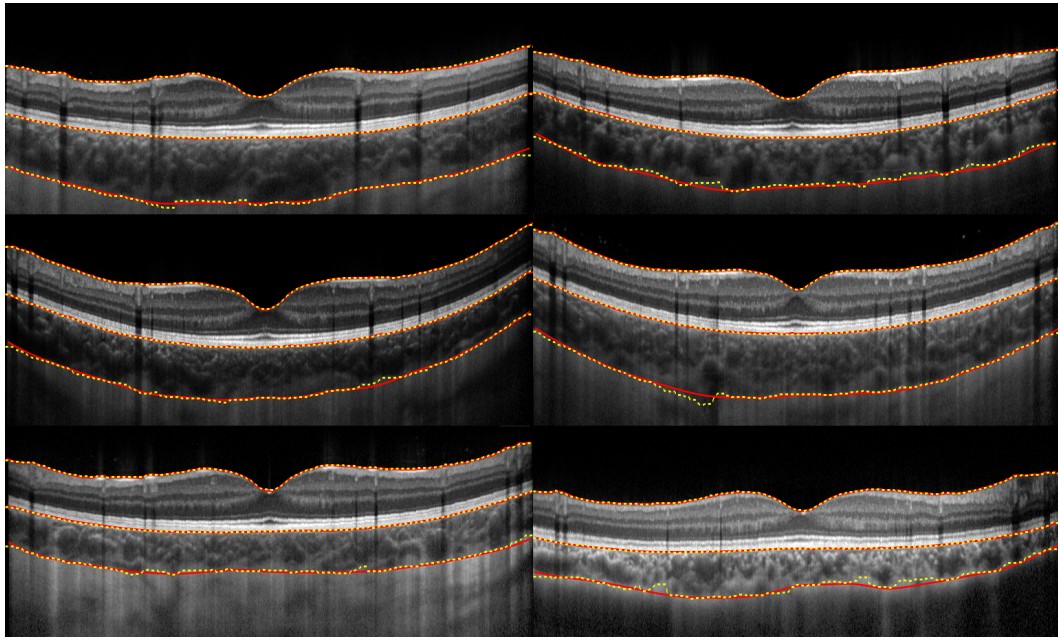

Figure 2: Figure 2. Six B-scans representing different examples from different participants, with typical variations in overall thickness and contrast. Manual segmentation is shown in red-solid and automatic is shown in yellow-dotted. The rectangular 31x61 network with standard intensity image was used to generate these results.

class, so graph-search is needed to extract the boundary. The performance of the methods seems to be strongly connected with the way the graph-search is performed (data not shown here).

Table 4. The difference in boundary position for each network architecture (proposed network versus fully convolution network) and the manual observer for the entire dataset. The results are reported as the mean value and (standard deviation) in pixel units. MAE = mean absolute error, ME = mean error.

|  | CNN-GS (31x61) | | FCNN (ReLayNet) | |
|---|---|---|---|---|
|  | MAE | ME | MAE | ME |
| ILM | 0.62 (2.08) | -0.15 (2.16) | 0.86 (2.12) | 0.42 (2.25) |
| RPE | 0.52 (0.53) | 0.01 (0.74) | 0.83 (1.62) | 0.39 (1.78) |
| CSI | 3.24 (4.83) | 0.95 (5.74) | 3.95 (7.74) | -0.29 (8.68) |

## 4  Conclusions

A convolutional neural network (CNN) and graph-search method was proposed for the segmentation of the total retinal and choroidal thickness. In this work, we have shown that the CNN architecture can have a significant impact on the performance, particularly the patch size used for the CNN training. The different tested architectures had a particularly strong impact on the chorio-scleral boundary segmentation. Overall, the rectangular architecture (elongated along the A-scan), which takes advantage of the rich feature details along the axial direction, significantly improved the performance and the detection of the different boundaries. The result for this network showed very close agreement (Figure 2) between the automatic and manual segmentation methods, which suggests that the proposed method provides robust detection of the retinal and choroidal boundaries of interest. As anticipated, the chorio-scleral boundary, which does not have a well-defined boundary, shows more variability than retinal boundaries.

We also evaluated the effect of different input images (standard, contrast enhanced and dual) on network performance. However, the input-image had minimal effect on the tested dataset. Given that a number of previous studies have used a fully convolution network for the segmentation of images,

here the proposed network was assessed against a fully convolution network, and the proposed network provided similar results and outperformed the FCNN in some metrics. However, it is worth noting that the graph-search used to extract the boundaries may also negatively contribute to the results. The use of more sophisticated methods to extract the boundaries for the FCNN network should be considered in the future [16].

In summary, the proposed method provides encouraging results for the automated segmentation of total retinal and choroidal thickness in OCT images, showing a good agreement with an experienced human observer. This work may support the future development of CNN methods for automated OCT boundary segmentation.

### Acknowledgments

We gratefully acknowledge support from the NVIDIA corporation for the donation of GPUs used in this work. Rebecca L. Cooper 2018 Project Grant (DAC); Telethon – Perth Children's Hospital Research Fund (DAC).

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
