# OpenReview forum: "Use of convolutional neural networks for the automatic segmentation of total retinal and choroidal thickness in OCT images"
_MIDL.amsterdam/2018/Conference — Submitted to MIDL 2018_

### Review · AnonReviewer2 · 2018-05-04
**a well-written paper that uses deep learning as a pre-processing module followed by graph-search algorithm**

**Rating:** 3
**Confidence:** 3

**Review:**

The paper uses the graph-search algorithm for segmenting OCT images.  Instead of using custom-made response function in the graph-search, the paper uses deep learning to derive a more responsive map. As a result, it obtains better results than simply using FCN as reported by their experiment on over 100 images.

Though it is probably not done before, the proposed approach is of no surprise. Therefore, the overall novelty is rather marginal.

From Figure 2, it is clear that the obtained results are still quite 'noisy' when compared with ground truth. Is there a way to encourage some smoothness in the final output? Also, deep learning is used as a precursor, is it possible to perform the learning in an end-to-end fashion, that is, the derivative is propagated from the graph-search algorithm?



**Special Issue:**

No

---

> ### Comment · ~David_Alonso-Caneiro1 · 2018-05-13
> **Thanks for the review, please find our comments below.**
>
> REVIEWER COMMENT: From Figure 2, it is clear that the obtained results are still quite 'noisy' when compared with ground truth. Is there a way to encourage some smoothness in the final output? Also, deep learning is used as a precursor, is it possible to perform the learning in an end-to-end fashion, that is, the derivative is propagated from the graph-search algorithm?
>
> ANSWER: The review is correct that the boundary for the choroid (CSI) is more variable, particularly in comparison to the other two boundaries (RPE and ILM). This is partly due to the way the “ground truth” data was manually segmented. The observer was instructed to select about 20 points on the boundary and a smooth function (spline fit) was used to connect those points. Thus, the boundary is inherently smooth. We wanted to be fair in the comparison and thus post-processing of the boundary position (smoothing) was not performed. Although we agree with the review that it has the potential to improve the results. Yet to be fair we have decided not to perform this task.
>
> We also agree with the reviewer that an end-to-end method is our future goal, thus skipping the graph-search step. In the conclusion, we have already suggested that the graph-search may have a potential negative impact on the performance and using a neural network (similar to He et al work) may provide better performance. Yet the work presented in here sets the basis for the development on the first stage of a future end-to-end approach.
>
>  [16] Y. He, A. Carass, B. N. Jedynak, S. D. Solomon, S. Saidha, P. A. Calabresi, & J. L. Prince, "Topology guaranteed segmentation of the human retina from OCT using convolutional neural networks". arXiv preprint arXiv:1803.05120 (2018).

---

### Review · AnonReviewer1 · 2018-05-07
**TITLE: Use of convolutional neural networks for the automatic segmentation of total retinal and choroidal thickness in OCT images**

**Rating:** 2
**Confidence:** 2

**Review:**

The paper describes a methodology based on Deep Neural Networks to automatically segment the outmost(and innermost) boundaries of the retinal layers, and the boundaries of the chorio-scleral interface in retinal Optical Coherence Tomography imaging. The authors compare the performance of some network architectures and study the impact of common preprocessing operations in the automatic segmentation task.  The results in the manuscript suggest that a convolutional network architecture with rectangular shaped filters (the filters have different sizes across the dimensions) might perform better than usual filters (with the same size across the dimensions) for the automatic segmentation task. Also, the authors did not find significant improvement when using common preprocessing operations in the retinal segmentation task. The number of patients and OCT images in the study is significant (Close to 100 patients). However, the contribution of the paper is not clear (or it is marginal) and there are several inconsistencies. After reading the paper I have the following concerns:

* Comment 1
In Section 4. Conclusions, the authors claim to have proposed a CNN and a graph-search method. However, the graph-search method is not described in the manuscript.

* Comment 2
In Section 3. Results, the performance gain for one of the architectures is attributed to the rectangular shape of the convolutional filters in the neural network. However, the difference in performance might not be caused only (or mainly) by the difference in the convolutional filters size or shape. Specifically, the architecture that uses image patches with size 31x61 has a total of 5 convolutional layers, while the architecture that uses squared image patches has four (4) or three (3) convolutional layers. There are many different factors among the architectures: Number of convolutional layers, patches size, the filter size in the convolutional layers, etc. The experiments do not allow to conclude which of these factors is responsible for the performance gain. Additionally, the criteria to select/design the neural architectures used in the experiments is not explained.

* Comment 3
  The title wording is weird and seems to be too general. Specifically 'automatic segmentation of ... choroidal thickness' sounds inaccurate.

* Comment 4
- How many B-scans does each patient has? How were the B-scans from the OCT volumes selected for extraction? Do they correspond to the central B-scan?

* Minor comments
There are several typos and grammar errors in the manuscript. Among them:
- 'centred'
- 'more standard image analysis procedures'
- 'adapts the patch size to better capture the richness of the information'

**Special Issue:**

No

---

> ### Comment · ~David_Alonso-Caneiro1 · 2018-05-13
> **Thanks for the review, please find our comments below.**
>
> REVIEWER COMMENT: Comment 1 In Section 4. Conclusions, the authors claim to have proposed a CNN and a graph-search method. However, the graph-search method is not described in the manuscript.
>
> ANSWER: We agree with the reviewer that more details regarding the graph-search is required and for completeness the revised version of the paper will include a more comprehensive description of the details regarding the graph-search.
>
> REVIEWER COMMENT: Comment 2 In Section 3. Results, the performance gain for one of the architectures is attributed to the rectangular shape of the convolutional filters in the neural network. However, the difference in performance might not be caused only (or mainly) by the difference in the convolutional filters size or shape. Specifically, the architecture that uses image patches with size 31x61 has a total of 5 convolutional layers, while the architecture that uses squared image patches has four (4) or three (3) convolutional layers. There are many different factors among the architectures: Number of convolutional layers, patches size, the filter size in the convolutional layers, etc. The experiments do not allow to conclude which of these factors is responsible for the performance gain. Additionally, the criteria to select/design the neural architectures used in the experiments is not explained.
>
> ANSWER: It is worth noting that the all three networks were designed to have minimal differences in terms of their internal parameters, with the primary aim being to evaluate the impact of patch size rather than the network architecture. However, the reviewer is correct that due to the different architectures for each of the patch sizes, their internal architectures are slightly different, so the performance can be also be affected by the different number of internal parameters.
>
> However, we believe that the improvement that we observed for the patch size (extra information on the patch) is the main driver of the improved performance rather than the architecture. We have run an additional simulation to further highlight this. We have used the larger 65x65 network but masking (i.e. setting to zero) the input patch to simulate the reduction of information that we observed in a rectangular patch (31x61) and we compared the performance.
>
> Performance of the 65x65 network using the entire patch (as in the paper)
> MAE = mean absolute error, ME = mean error
> ------ MAE (SD)     ME (SD)   [pixel units]
> ILM   1.64 (4.68)    -0.70 (4.92)
> RPE  1.02 (2.19)    -0.07 (2.41)
> CSI 10.47 (32.32)  -2.44 (33.87)
>
> Performance of the 65x65 network using the masking the patch to zero to simulate a 31x61
> ------ MAE (SD)     ME (SD)   [pixel units]
> ILM   1.43 (4.63)    -0.69 (4.80)
> RPE  1.08 (2.17)    -0.08 (2.42)
> CSI  6.57 (18.44)    1.42 (19.50)
>
> It is clear for the choroidal boundary (CSI) that the performance of the method improves by reducing the amount of information (masking the input) used to train the network. Given that this analysis effectively alters the patch size while keeping the other network parameters constant, this result highlights the relative importance of patch size in the improved performance of delineating the choroidal boundary that is reported in the paper. However, we agree with the reviewer that the conclusions of the paper should be adjusted to reflect the potential impact of the different parameter on the performance. Also future work should look into ways to reduce the differences between architectures to isolate the effect of the patch size.
>
> REVIEWER COMMENT: Comment 3 The title wording is weird and seems to be too general. Specifically 'automatic segmentation of ... choroidal thickness' sounds inaccurate.
>
> ANSWER: We agree that in the manuscript we have not reported the thickness performance and have focused on the boundary error. Although related, it may be best to substitute the word “thickness” for “boundaries”. So the title in the revised version will be “Use of convolutional neural networks for the automatic segmentation of total retinal and choroidal boundaries in OCT images”
>
> REVIEWER COMMENT: Comment 4 - How many B-scans does each patient has? How were the B-scans from the OCT volumes selected for extraction? Do they correspond to the central B-scan?
>
> ANSWER: We agree with the reviewer that more details are needed in this section. Each participants had 2 series of 6 high resolution foveal centered radial OCT scan lines acquired using the instrument's Enhanced Depth Imaging (EDI) mode (with each radial scan line separated by 30 degrees). Participants contributed at most 2 scans for training. Training scans were chosen semi-randomly, with an initial random sampling, then a correction applied starting with the first participant until all scan orientations were balanced to ensure the same number of radial locations were used for training. Only participants with all 6 scans who had not contributed to training were selected for testing.

---

### Review · AnonReviewer3 · 2018-05-09
**Convolutional neural networks (CNNs) are used to calculate the probability of boundary locations in OCT images. The effect of patch size, network architecture and input image pre-processing on the CNN performance and subsequent layer segmentation are evaluated.**

**Rating:** 2
**Confidence:** 2

**Review:**

Summary
In this paper, convolutional neural networks (CNNs) are used to calculate the probability of boundary locations in OCT images. The CNN, trained using image patches centered around the boundary of interest, provides a per-layer probability map that marks the most likely predicted location of the boundaries. This map is subsequently traced using a graph-search approach to segment the boundaries. The authors base their approach on a well-established approach introduced by Fang et al [10] and show its applicability on Choroidal images. This work may be regarded as verification of the applicability of recent methods to Choroidal Segmentation.
The effect of patch size, network architecture and input image pre-processing on the CNN performance and subsequent layer segmentation are evaluated.

Detailed review
Methodology: some detail is lacking, including the selection method for training pathces, and the loss function used in the training phase.
The authors compare 3 different patch sizes and conclude that a rectangular patch is best suited for the task of generating probability maps. The authors mention that the patch size has the most impact on performance. Note that in addition to the patch size, additional variability is present in the network internal parameters (Filter sizes, Size of FC layer, position of Activiation RELU etc) making the statement that patch size was the main factor for improved performance less substantiated.

The authors reference previous works (e.g. X. Sui, [6], and [19]). These works should be discussed in reference to the current work – and probably compared quantitatively as well. In particular, Paper [6] - "Retinal layers segmentation using Fully Convolutional Network in OCT images",utilized a U-net based architecture for retinal layer segmentation and showed superior performance to a patch based method by great margins. Thus, the article reached a different conclusion to the one presented in the current work. This issue should be address in the Discussion
.
Authors evaluated the effect of different input images (standard, contrast enhanced and dual) and although this did not show improved performance, we find this comparison important.

Pros	-	Validation of a well-known approach to newly curated dataset of Choroidal images
-	Comparison of different input images.
-	Quality of the work is good
Cons	-	Lack of detail on Loss function, training patch selection method and balancing approach which is a crucial part in patch based training.
-	Graph search part not described or explored in depth.
-	Conclusion as to patch size effect on performance is not fully substantiated.
-	 Lacking comparison to previous work focusing on Choroidal Segmentation


**Special Issue:**

No

---

> ### Comment · ~David_Alonso-Caneiro1 · 2018-05-13
> **Thank you for the review, please find our comments below (1/2).**
>
> REVIEWER COMMENT: Methodology: some detail is lacking, including the selection method for training pathces, and the loss function used in the training phase.
>
> ANSWER: In the original manuscript, the details regarding loss functions were not included, in the revised version it will be clarified that the loss function was cross-entropy.
> Further details will also be provided regarding the rationale for the patch size selected in this work. Since we based our work on the previous study by Fang et al (2017) we used 33x33patch size as a reference. In our preliminary work, we observed that using this patch size resulted in some boundaries being easily confused with neighbouring boundaries. Thus, we hypothesized that an increase in patch size may improve the segmentation performance. We also hypothesized that increasing the horizontal patch size would have negligible effect on performance while compared to the vertical. Thus we tested the rectangular patch of 31x61 pixels as well as a second larger network (65x65 pixels patch size).
>
> L. Fang, D. Cunefare, C. Wang, R. H. Guymer, S. Li, & S. Farsiu, "Automatic segmentation of nine retinal layer boundaries in OCT images of non-exudative AMD patients using deep learning and graph search," Biomed. Opt. Express 8(5), 2732–2744 (2017).
>
> REVIEWER COMMENT: The authors compare 3 different patch sizes and conclude that a rectangular patch is best suited for the task of generating probability maps. The authors mention that the patch size has the most impact on performance. Note that in addition to the patch size, additional variability is present in the network internal parameters (Filter sizes, Size of FC layer, position of Activiation RELU etc) making the statement that patch size was the main factor for improved performance less substantiated.
>
> ANSWER: It is worth noting that the all three networks were designed to have minimal differences in terms of their internal parameters, with the primary aim being to evaluate the impact of patch size rather than the network architecture. However, the reviewer is correct that due to the different architectures for each of the patch sizes, their internal architectures are slightly different, so the performance can be also be affected by the different number of internal parameters.
>
> However, we believe that the improvement that we observed for the patch size (extra information on the patch) is the main driver of the improved performance rather than the architecture. We have run an additional simulation to further highlight this. We have used the larger 65x65 network but masking (i.e. setting to zero) the input patch to simulate the reduction of information that we observed in a rectangular patch (31x61) and we compared the performance.
>
> Performance of the 65x65 network using the entire patch (as in the paper)
> MAE = mean absolute error, ME = mean error
> ------ MAE (SD)     ME (SD)   [pixel units]
> ILM   1.64 (4.68)    -0.70 (4.92)
> RPE  1.02 (2.19)    -0.07 (2.41)
> CSI 10.47 (32.32)  -2.44 (33.87)
>
> Performance of the 65x65 network using the masking the patch to zero to simulate a 31x61
> ------ MAE (SD)     ME (SD)   [pixel units]
> ILM   1.43 (4.63)    -0.69 (4.80)
> RPE  1.08 (2.17)    -0.08 (2.42)
> CSI  6.57 (18.44)    1.42 (19.50)
>
> It is clear for the choroidal boundary (CSI) that the performance of the method improves by reducing the amount of information (masking the input) used to train the network. Given that this analysis effectively alters the patch size while keeping the other network parameters constant, this result highlights the relative importance of patch size in the improved performance of delineating the choroidal boundary that is reported in the paper. However, we agree with the reviewer that the conclusions of the paper should be adjusted to reflect the potential impact of the different parameter on the performance. Also future work should look into ways to reduce the differences between architectures to isolate the effect of the patch size.

---

> ### Comment · ~David_Alonso-Caneiro1 · 2018-05-13
> **Thank you for the review, please find our comments below (2/2).**
>
> REVIEWER COMMENT: The authors reference previous works (e.g. X. Sui, [6], and [19]). These works should be discussed in reference to the current work – and probably compared quantitatively as well. In particular, Paper [6] - "Retinal layers segmentation using Fully Convolutional Network in OCT images",utilized a U-net based architecture for retinal layer segmentation and showed superior performance to a patch based method by great margins. Thus, the article reached a different conclusion to the one presented in the current work. This issue should be address in the Discussion .
>
> ANSWER: In table 4 we showed the comparison of the best proposed network against a fully convolution methods, and we stated that this network provides “similar results and outperformed the FCNN in some metrics”. The improvement was evidenced by the smaller standard deviation for the choroidal boundary error.  The use of other networks (such as U-net) for further comparison is a good recommendation and we thank the review for this suggestion. In the original manuscript we chose ReLayNet, since ReLayNet is reported to have a superior performance that U-net for retinal segmentation, based on Roy et al. 2017 paper. Yet this comparison should be included in the future, since U-net is a commonly used architecture for this segmentation task.
>
> A. G. Roy, S. Conjeti, S. P. K. Karri, D. Sheet, A. Katouzian, C. Wachinger, & N. Navab, "ReLayNet: retinal layer and fluid segmentation of macular optical coherence tomography using fully convolutional networks," Biomed. Opt. Express 8 (8), 3627–3642 (2017).
>
> REVIEWER COMMENT: Lack of detail on Loss function, training patch selection method and balancing approach which is a crucial part in patch based training.
>
> ANSWER: The reviewer is correct that more details are needed in the paper. In the revised version it will be highlighted that the loss function was cross-entropy. Similarly, it will also be clarified that for the training patch selection all columns within the image with all three layers were used. The boundary classes (ILM, RPE, CSI) were sampled at the centre of the boundary while one randomly sampled background was also selected per image column (i.e. A-scan). The location of the background patch was restricted to be no more than half the patch size above or below the outermost layers (ILM and CSI). As such all classes are balanced equally, and scans and participants chosen such that all orientations appeared in equal amounts. There was also no overlap of participant between test and training sets.
>
> REVIEWER COMMENT: Graph search part not described or explored in depth.
>
> ANSWER: Although we have not changed the graph-search from our previous published work, it is true that for completeness the revised version of the paper should include more details regarding the graph-search.
>
> REVIEWER COMMENT: Conclusion as to patch size effect on performance is not fully substantiated.
>
> ANSWER: We hope the extra comments provided above will clarify this aspect of the study.
>
> REVIEWER COMMENT: Lacking comparison to previous work focusing on Choroidal Segmentation
>
> ANSWER: The reviewer is correct and this should be included in future revision of the paper.

---

### Decision · Program_Chairs · 2018-05-15
**Paper23 Acceptance Decision**

Reject